# Co-evolution Transformer for Protein Contact Prediction

**He Zhang**[*][†]
Xi'an Jiaotong University
mao736488798@stu.xjtu.edu.cn

**Fusong Ju**[†]
Microsoft Research Asia
fusongju@microsoft.com

**Jianwei Zhu**
Microsoft Research Asia
jianwzhu@microsoft.com

**Liang He**
Microsoft Research Asia
lihe@microsoft.com

**Bin Shao**[‡]
Microsoft Research Asia
binshao@microsoft.com

**Nanning Zheng**[‡]
Xi'an Jiaotong University
nnzheng@mail.xjtu.edu.cn

**Tie-Yan Liu**
Microsoft Research Asia
tyliu@microsoft.com

## Abstract

Proteins are the main machinery of life and protein functions are largely determined by their 3D structures. The measurement of the pairwise proximity between amino acids of a protein, known as inter-residue contact map, well characterizes the structural information of a protein. Protein contact prediction (PCP) is an essential building block of many protein structure related applications. The prevalent approach to contact prediction is based on estimating the inter-residue contacts using hand-crafted coevolutionary features derived from multiple sequence alignments (MSAs). To mitigate the information loss caused by hand-crafted features, some recently proposed methods try to learn residue co-evolutions directly from MSAs. These methods generally derive coevolutionary features by aggregating the learned residue representations from individual sequences with equal weights, which is inconsistent with the premise that residue co-evolutions are a reflection of collective covariation patterns of numerous homologous proteins. Moreover, non-homologous residues and gaps commonly exist in MSAs. By aggregating features from all homologs equally, the non-homologous information may cause misestimation of the residue co-evolutions. To overcome these issues, we propose an attention-based architecture, Co-evolution Transformer (CoT), for PCP. CoT jointly considers the information from all homologous sequences in the MSA to better capture global coevolutionary patterns. To mitigate the influence of the non-homologous information, CoT selectively aggregates the features from different homologs by assigning smaller weights to non-homologous sequences or residue pairs. Extensive experiments on two rigorous benchmark datasets demonstrate the effectiveness of CoT. In particular, CoT achieves a $51.6\%$ top-L long-range precision score for the Free Modeling (FM) domains on the CASP14 benchmark, which outperforms the winner group of CASP14 contact prediction challenge by $9.8\%$.

---

[*]Work done during an internship at Microsoft Research.
[†]Equal contribution.
[‡]Corresponding authors.

35th Conference on Neural Information Processing Systems (NeurIPS 2021).

# 1 Introduction

Most *in vivo* biological processes are carried out by proteins, whose functions are mainly determined by their 3D structures [1]. Structural information is crucial for understanding the functions of a protein. The inter-residue contact map which measures the pairwise proximity between all amino acid pairs well characterizes the structural information of a protein. Protein contact prediction (PCP) is an important building block of many protein structure related applications, such as protein structure prediction [2, 3], protein complex assembly [4], and protein design [5]. It is so important that it is one of the challenges of Critical Assessment of protein Structure Prediction (CASP), which is the "world championship" of the computational structural biology field.

The prevalent PCP approaches were built atop the co-evolution principle that the spatially proximate residues tend to co-evolve to maintain the functions of a protein [6–8]. The existing PCP methods generally infer pairwise coevolutionary patterns from MSAs. Among them, direct coupling analysis (DCA) [9] techniques are widely used to obtain coevolutionary features by fitting Potts models or calculating precision matrix [10, 11]. Subsequently, a variety of deep-learning based methods were proposed to leverage the DCA-based features to estimate inter-residue contacts [3, 12–15]. However, DCA techniques only consider single-residue and pairwise statistics of the sequences, ignoring high-order interactions among the residues within a sequence.

To mitigate the information loss caused by hand-crafted features (e.g., DCA-based features), some recently proposed methods try to learn residue co-evolutions directly from MSAs [2, 16, 17]. For example, the SOTA of them, CopulaNet [2] learns residue representations from individual sequences and then aggregates these features with equal weights to derive residue co-evolutions. This causes two issues: 1) Inferring residue representations from individual sequences independently is inconsistent with the premise that residue co-evolutions are a reflection of collective covariation patterns of numerous homologs [18]; 2) Assigning equal weights to the features coming from different homologs ignores the fact that, the quality of homologs varies a lot because of the existence of non-homologous residues and gaps [19, 20].

In this paper, we propose an attention-based architecture, Co-evolution Transformer (CoT), to address or mitigate the issues discussed above. The core component of CoT is the *co-evolution attention* (CoA) module. Different from the previous methods that focus on extracting information from individual sequences, CoA is capable of incorporating the residue co-evolution patterns derived from all homologous sequences into an attention function to learn residue representations. Moreover, the CoA learns to automatically weight the residue representations learned from different homologs, and then selectively aggregates the features to construct the co-evolution attention map. This design mitigates the influence of non-homologous information.

CoT dramatically outperforms the baseline methods on two rigorous benchmarks CASP14 [21] and CAMEO (Continuous Automated Model EvaluatiOn) [22]. In particular, CoT achieves a $51.6\%$ top-L long-range precision score for the Free Modeling (FM) domains on the CASP14 benchmark, which outperforms the winner group of CASP14 contact prediction challenge by $9.8\%$. Our code will be released at https://github.com/microsoft/ProteinFolding/tree/main/coevolution_transformer.

# 2 Related Work

Protein contact prediction is a binary classification task for amino acid residue pairs. A residue pair is called a contact if their distance is less than or equal to a distance threshold, typically 8Å, i.e., $8 \times 10^{-10}$m.

As widely acknowledged, co-evolution information is closely correlated to the contacts. In order to extract the co-evolution patterns for residue pairs, multiple sequence alignments (MSAs) are generated from raw protein sequences. For a target protein sequence, a generated alignment consists of multiple sequences with each corresponding position being an aligned residue or a gap (annotated by a dash), and these aligned sequences as a whole are called a multiple sequence alignment. Many protein databases as well as various search schemes have been proposed to generate MSAs efficiently [19, 20]. There are roughly three categories of methods for predicting contacts from MSAs, namely, unsupervised methods, supervised methods, and pre-training based methods.

**Unsupervised Methods**    To quantify the strength of the direct relation between the residue pairs of a protein sequence while excluding effects from other residues, many statistical modeling methods based on direct coupling analysis (DCA) have been proposed to fit Potts models [10] or precision matrix [11] to MSAs, e.g., mean-field DCA [9], sparse inverse covariance [11] and pseudo-likelihood maximization [10, 23, 24], to name a few. These methods further exploit dedicated scores based on DCA for contact prediction. However, they only consider single-residue and pairwise statistics of the sequences, thus fail to capture high-order interactions among the multiple residues within a sequence.

**Supervised Methods**    By taking DCA-derived scores as features, deep neural networks based supervised methods significantly outperform the unsupervised methods [3, 12–15]. However, the information lost by the DCA-based features from the sequences is still not recoverable. To mitigate this issue, several models are proposed to learn residue co-evolution information directly from the sequences in the MSAs [2, 16, 17, 25]. Among them, CopulaNet [2], the SOTA of the CASP13 benchmark, derives coevolutionary features differentially by aggregating the learned residue representations from the sequences. However, although they are capable of modeling the high-order interactions among the multiple residues within single sequence, the global information carried by the MSAs is ignored because they still model the protein sequences independently. AlphaFold2 [25] is claimed to be modeling the full MSAs, achieving an amazing performance on the CASP14 structure prediction task. Although its performance on the structure prediction task is remarkable, they did not participate in the CASP14 PCP task meanwhile no further details of their methods are publicly available.

**Pre-training Based Methods**    Following the *pre-train and fine-tune* paradigm, pre-trained language models are adapted to representation learning for single protein sequences from the unlabeled data [26–31]. Many of them take contact prediction as an important downstream task to validate their performance. While these methods show another solution to this task, they are still at an early stage thus cannot achieve comparable performance to the SOTA approaches currently. To further improve the performance, a pre-trained language model named MSA Transformer is proposed to learn a better MSA representation directly [29]. MSA Transformer did solid work on learning co-evolution information from unlabeled MSAs. However, non-homologous subsequences are inevitably introduced during the learning process.

To exploit the co-evolution information from the full MSAs effectively, our proposed CoT model is built upon *co-evolution attention*, a novel attention mechanism dedicated to incorporate the co-evolution information directly from MSAs in a supervised way.

## 3    Co-evolution Transformer

Co-evolution Transformer (CoT) is constructed by stacking several repeated CoT layers. Each CoT layer is composed of two attention modules, i.e., a *co-evolution attention* (CoA) module and a *self-attention* module, as shown in Figure 1.

Given a prepared MSA, the stacked CoT layers are used to learn the residue representations. Residue co-evolutions are derived from the representations of the final layer and further employed to estimate the inter-residue contacts.

In this section, we first start with a brief introduction to the vanilla Transformer Encoder, followed by the detailed descriptions of each CoT component. To better illustrate the co-evolution attention mechanism, the proposed co-evolution attention mechanism and the self-attention mechanism are discussed in the end.

### 3.1    Vanilla Transformer Encoder

Transformer is a network architecture built on the attention mechanism for machine translation. The Transformer encoder is widely adopted for machine learning tasks due to its excellent feature extraction capability when modeling long-distance interactions in sequences [32–34]. Each Transformer encoder layer consists of two modules, i.e., a *multi-head self-attention* (MHSA) module and a *position-wise fully connected feed-forward* (FFN) module. To connect these two modules, residual

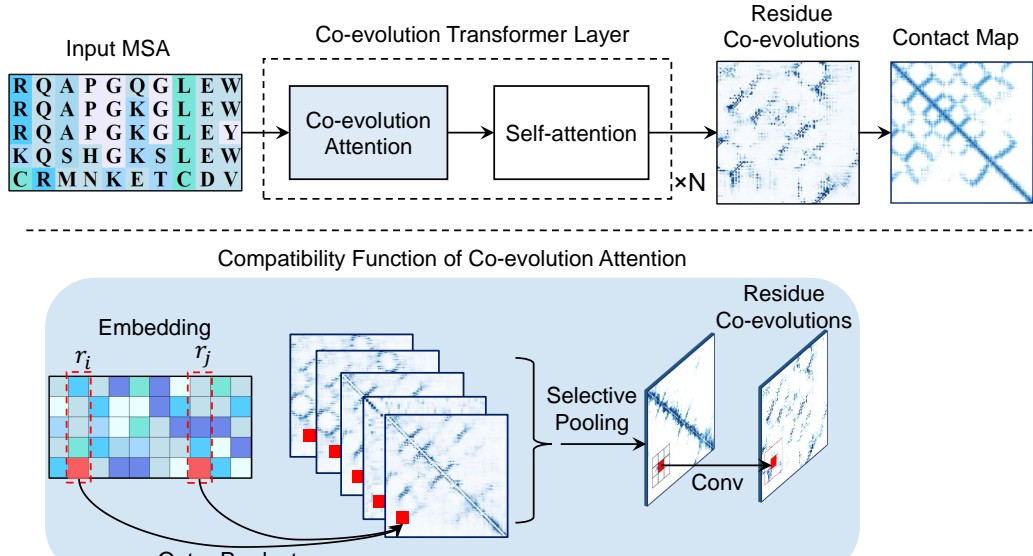

Figure 1: An Overview of Co-evolution Transformer. Given an input MSA, several CoT layers are stacked to learn residue representations, which are then used to derive residue co-evolutions to estimate inter-residue contacts. Each CoT layer consists of a co-evolution Attention (CoA) module and a self-attention module.

connections [35] and layer normalization (LAYERNORM) [36] are applied as below:

$$x = \text{LAYERNORM}(x + \text{MHSA}(x)), \tag{1}$$
$$x = \text{LAYERNORM}(x + \text{FFN}(x)), \tag{2}$$

The multi-head self-attention module is also adopted by the CoT layer as a component to learn the residue representations of different sequences in the MSA, however, iterating through the columns instead of rows/sequences.

## 3.2 Co-evolution Attention Module

For a target protein sequence and its MSA, the goal of the Co-evolution Attention Module is to leverage the whole MSA to derive pairwise inter-residue interaction features, namely residue co-evolutions. Intuitively, the residue co-evolutions are analogous to the covariance matrix in mathematics, depicting the correlations between any two residue positions (two columns in the MSA). Then, these features are used as attention maps to guide representation learning of each sequence in the MSA. To achieve this goal, a CoA module employs two consequent submodules, i.e. *co-evolution aggregation* and *co-evolution enhancement*. The co-evolution aggregation submodule is designed to generate the coevolutionary features by aggregating pairwise interactions from all homologs in the MSA, while the co-evolution enhancement submodule further enhances the coevolutionary features and derives the final co-evolution attention. The overview of the CoA module is illustrated in Figure 2.

Given the target protein sequence $(r_1, r_2 \ldots r_L)$, where $L$ is the length of the sequence, the corresponding representation of its MSA is denoted as $X \in \mathbb{R}^{K \times L \times d_{model}}$, where $K$ is the number of homologous sequences and $d_{model}$ is the hidden dimension of the residue representation. A detailed representation schema of the MSA is described in Appendix.

For the $k$-th sequence in the MSA, the overall procedure of the CoA module in each layer can be summarized as follows:

$$X^k = \text{LAYERNORM}(X^k + \text{COATTN}(X)), \tag{3}$$
$$X^k = \text{LAYERNORM}(X^k + \text{FFN}(X^k)) \tag{4}$$

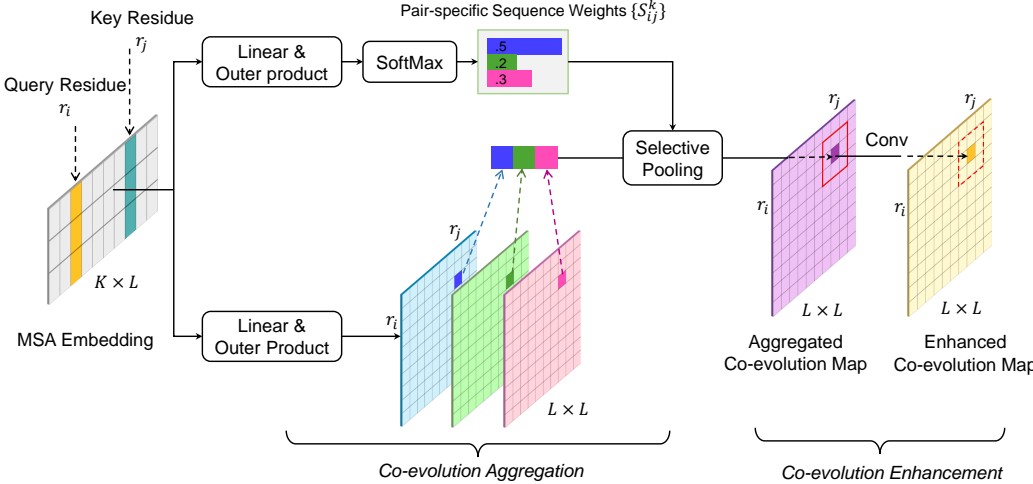

Figure 2: Schema of Co-evolution Attention. Given a residue pair $\langle r_i, r_j \rangle$, we first learn $K$ covariance features and a set of pair-specific sequence weights $S_{ij}^k$, $k = 1 \cdots K$. The weights are used to selectively aggregate the $K$ features. Finally, the aggregated feature is fed to a convolutional module to obtain the co-evolution attention map. For better visualization, we omit the feature channels and the details about how co-evolution attention attend to residue embeddings.

where $k \in [1 \dots K]$ and COATTN is defined as:

$$\text{COATTN}(X) = \text{CONCAT}(\text{head}_1, \dots, \text{head}_H), \tag{5}$$

$$\text{head}_h = \text{ATTN}_h(X, A) \, X_h^k \, W_h, \tag{6}$$

where $h \in [1 \dots H]$ is the head index, $A \in \mathbb{R}^{L \times L \times d_{co}}$ is the co-evolution feature map generated with $d_{co}$ as its feature dimension, $\text{ATTN}_h(X, A) \in \mathbb{R}^{L \times L}$ is the $h$-th co-evolution attention head, $X_h^k \in \mathbb{R}^{L \times d_v}$ is the $h$-th segment of $X^k$, and $W_h \in \mathbb{R}^{d_v \times d_v}$ are learnable weights ($d_v = d_{model}/H$). Note that the co-evolution attention $\text{ATTN}_h(X, A)$ is shared by all the $K$ homologous sequences within the same head.

**Submodule 1: Co-evolution Aggregation**   To depict inter-residue interactions, the co-evolution aggregation submodule exploits outer product on the residue representations. A selective pooling operation is then applied to the aggregated outer products from all homologous sequences, the disturbance of non-homologous information can be greatly reduced by this weighting mechanism.

Two tensors $P, Q \in \mathbf{R}^{K \times L \times d_{model}}$, are generated from $X$ by separate linear projections. For the residue pair $\langle r_i, r_j \rangle$ in the $k$-th sequence of the MSA, the pair co-evolution features $A_{ij}$ are calculated by:

$$A_{ij} = \text{PROJ}\left( \sum_{k=1}^{K} S_{ij}^k \odot (P_i^k \otimes P_j^k) \right), \tag{7}$$

and

$$S_{ij}^k = \frac{1}{Z} \exp(Q_i^k \otimes Q_j^k), \tag{8}$$

where $\otimes$ is the outer product operator, $\odot$ is the element-wise multiplication operator, PROJ is a flatten operation followed by a linear projection, converting the aggregated pair co-evolution features to $A \in \mathbb{R}^{L \times L \times d_{co}}$, and $Z$ stands for the normalization factor. The weight for the pair representation is denoted as $S_{ij}^k$, where $S_{ij}^k \in \mathbb{R}^{d_{model} \times d_{model}}$ whose elements fall in the range of $[0, 1]$. The weights are normalized over the $k$ sequences, thus we have $\sum_k S_{ij}^k = \mathbf{1}$.

**Submodule 2: Co-evolution Enhancement**   To model the high-order interactions among multiple residues, the co-evolution map $A$ is concatenated with $A'$ from the previous CoT layer, the values are

fed into a convolutional module (CONV) to generate an enhanced co-evolution map by:

$$A = \text{CONV}([A'; A]). \tag{9}$$

In practice, we adopt a ResNet [35] as the convolution module. Note that we simply assign $A'$ an all-zero tensor for the first layer.

The $h$-th co-evolution attention head is projected from $A$ by

$$\text{ATTN}_h(X, A) = \text{SOFTMAX}(A\, M_h), \tag{10}$$

where $M_h \in \mathbb{R}^{d_{co} \times 1}$ are learnable weights.

### 3.3 Why Co-evolution Attention

A straightforward way to model the co-evolution is by applying self-attention to individual sequences in the MSA. In this case, for a residue pair $\langle r_i,\, r_j \rangle$ in the $k$-th homolog, self-attention measures their compatibility (attention weight, $\mathcal{A}_{ij}^k$) by:

$$\mathcal{A}_{ij}^k \propto \text{SOFTMAX}\{(U X_i^k)^\top (V X_j^k)\}, \tag{11}$$

where $U, V$ are learnable weights.

In comparison, the attention weight of COA, calculated by the co-evolution aggregation module (AGGREGATE) and the co-evolution enhancement module (ENHANCE), can be summarized as:

$$A_{ij} = \text{AGGREGATE}(X_i \otimes X_j), \tag{12}$$
$$A_{ij} = \text{ENHANCE}(A_{ij}, \text{NEIGHBOR}(i,\, j)), \tag{13}$$
$$\mathcal{A}_{ij}^k \propto \text{SOFTMAX}\{A_{ij}\}, \tag{14}$$

Compare with self-attention, co-evolution attention is more expressive at three aspects: 1) COA leverages the global information from all the homologs instead of single homolog to derive the attention, which fits better to the residue co-evolution insight; 2) COA handles the non-homologous information naturally by the selective pooling operation during aggregation, providing a solution to a widely existed but inevitable dilemma for MSAs; 3) COA enhances the co-evolution signal by propagating information from the neighbors, making it easier to capture the high-order interactions among multiple residues.

## 4 Experimental Evaluation

We have conducted extensive experiments and analyses to evaluate the effectiveness of CoT.

### 4.1 Experiment Setup

**Benchmark**   Two standard benchmarks are used for the model evaluation in the conducted experiments, i.e., CASP14 and CAMEO. 1) CASP14 is the latest and most important benchmark for protein contact prediction [21]. This benchmark includes three kinds of protein domains, i.e., FM (22 domains), FM/TBM (14 domains), and TBM (50 domains), where a domain is a protein sequence prepared by the CASP organizers. The first protein among them was released on May. 18, 2020. 2) CAMEO is another benchmark to evaluate weekly-updated protein structure submissions continuously [22]. The proteins are classified into three categories: easy, medium, and hard. Among them, 176 hard targets, released in the last year (from 2020-04-17 to 2021-04-10), are selected for model evaluation.

**Dataset**   All models are trained on $96,167$ protein structures (chains) collected from PDB [37] (before Apr. 1, 2020), which are split into the train and validation sets ($95,667$ and $500$ proteins, respectively). For each protein sequence, we generate its MSA by searching UniRef30 (version 2020-02), UniRef90 (version 2020-02), BFD30 (version 2019-03), MGnify90 (version 2019-05) with HHblits (version 3.3.0) and HMMER (version 3.3.2). The details of the MSA generation procedure are described in Appendix.

Table 1: Comparison on CASP14 and CAMEO (*Precision@L* )

| Methods | CASP14 | | | CAMEO |
| | FM (22) | FM/TBM (14) | TBM (50) | Hard (176) |
|---|---|---|---|---|
| RaptorX [14] | 33.9 | 58.1 | 63.1 | 53.2 |
| trRosetta [15] | 31.3 | 57.6 | 61.1 | 50.1 |
| CopulaNet [2] | 38.5 | 62.2 | 65.5 | 56.5 |
| CoT-SA (ours) | 41.8 | 59.2 | 67.9 | 59.8 |
| CoT (ours) | **48.2** | **66.7** | **75.6** | **66.6** |

Table 2: Comparison on CASP14. Gr. 368, Gr. 488, and Gr. 010 are the results of the top-3 groups in the CASP14 challenge. CoT$^{\dagger}$ refers to the results of CoT with MSA selection.

| Method | FM (22) | | | FM/TBM (14) | | | TBM (50) | | |
| | $L$ | $L/2$ | $L/5$ | $L$ | $L/2$ | $L/5$ | $L$ | $L/2$ | $L/5$ |
|---|---|---|---|---|---|---|---|---|---|
| Gr. 368 | 41.8 | 55.7 | 66.6 | 64.5 | 78.6 | 87.4 | 73.1 | 87.1 | 94.5 |
| Gr. 488 | 40.4 | 52.9 | 65.0 | 63.6 | 78.8 | 88.5 | 72.0 | 86.9 | 93.7 |
| Gr. 010 | 39.6 | 53.4 | 63.8 | 61.5 | 77.0 | 86.8 | 66.1 | 80.9 | 89.5 |
| CoT$^{\dagger}$ (ours) | **51.6** | **68.2** | **79.9** | **66.8** | **82.2** | **90.5** | **77.9** | **91.0** | **96.1** |

**Evaluation**    Following the procedure of trRosetta [15], the contact prediction task is converted into a multi-class classification task. The inter-residue distance range is divided into 37 bins, i.e., $(0\,\text{Å}, 2.5\,\text{Å}], (2.5\,\text{Å}, 3.0\,\text{Å}], \cdots, (20.0\,\text{Å}, +\infty)$, while the models are trained with the bin labels. For contact, the summed probability value of the bins with distance less than $8\,\text{Å}$ are used as the final prediction.

**Metrics**    For the evaluation criterion, the prevalent metrics are employed, which are *Precision@L*, *Precision@L/2*, and *Precision@L/5* of long-range residue contacts, where *Precision@n* stands for the precision score for the top-$n$ pairs of the highest probability in the predicted contact map. Here, $L$ refers to the length of protein sequence and *long-range* means there are at least 23 other residues between these two residues in the sequence.

**Implementations**    Given an MSA, 256 sequences are randomly sampled and cropped by length 200. The CoT model is equipped with 6 CoT layers with hidden size as 128 and the attention head number as 8. All models are trained with Adam optimizer [38] via a cross-entropy loss for $100k$ iterations. The learning rate, the weight decay, and the batch size are set to $10^{-4}$, 0.01, and 16 respectively. The hyperparameters of the CoT model is selected according to the performance on the validation set, and a detailed comparison of different hyperparameter settings are summarized in Appendix. The total training cost of the CoT model is about 30 hours on 4 Tesla V100 GPU cards.

## 4.2  Evaluation Results

For the sake of fairness, CoT is compared with three SOTA methods for contact prediction, including RaptorX [14], trRosetta [15] and CopulaNet [2], on the identical MSAs by searching BFD30 using HHblits with default parameters. As shown in Table 1, CoT outperforms the baselines, while surpassing CopulaNet, the best of the SOTAs, by 9.7%, 4.5%, 10.1% and 10.1% for *Precision@L* scores on four kinds of targets, respectively. MSA Transformer [29] is not included in the comparison due to the different task settings (pretrained vs. supervised) and the unavailability of its supervised fine-tuning codebase. Nevertheless, we rerun the CoT model on the CASP13-FM dataset to compare with it, obtaining a 65.0% *Precision@L* score, which is better than 57.1% reported in MSA Transformer. A detailed discussion is described in Appendix.

Methods used by the groups participated in the CASP14 challenge are considered to the best for the benchmark due to their deep optimization towards MSA generation and model ensemble strategy. To further evaluate the performance of the proposed method, the top-3 winner groups/methods on

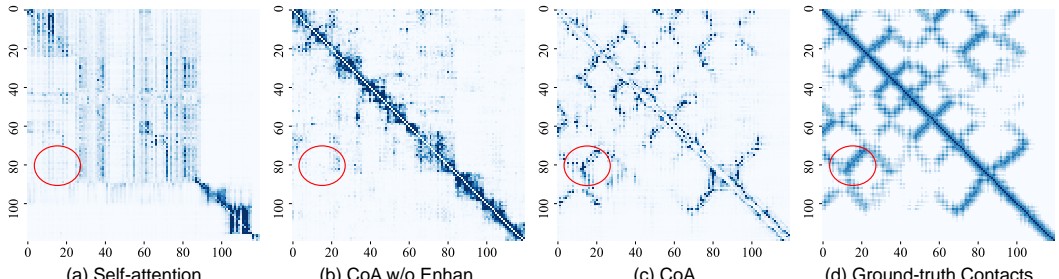

Figure 3: Comparison of the extracted attention maps for `4q2z_H`. (a) CoT-SA model; (b) CoT model w/o co-evolution enhancement; (c) CoT model; (d) Ground-truth contact map. The red circle covers typical long-range contacts.

Table 3: Ablations for CoA on CASP14 (*Precision@L*). AGGRE., ENHAN. and SA refer to the co-evolution aggregation submodule, the co-evolution enhancement submodule, and the self-attention module, respectively.

| | | | CASP14 | | | CAMEO |
|---|---|---|---|---|---|---|
| AGGRE. | ENHAN. | SA | FM (22) | FM/TBM (14) | TBM (50) | Hard (176) |
| Selective Pooling | ✓ | ✓ | 48.2 | 66.7 | 75.6 | 66.6 |
| Average Pooling | ✓ | ✓ | 42.7 | 62.2 | 73.6 | 64.2 |
| Selective Pooling | | ✓ | 41.6 | 61.2 | 70.3 | 61.8 |
| Selective Pooling | ✓ | | 46.4 | 66.9 | 74.8 | 66.3 |

CASP14 are compared with CoT. Different groups use different data sources to build their MSAs. To eliminate the variance of different data sources, the most confident prediction (the prediction with the highest probability score) of CoT on 12 MSAs from different databases with different search settings, denoted as CoT$^\dagger$, is selected as the final prediction.

As shown in Table 2, CoT$^\dagger$ outperforms the best method. On the hardest domains (CASP14-FM), CoT$^\dagger$ even increases 9.8%, 12.5%, 13.3% scores for all the three metrics, compared with Gr. 368, the best group on this benchmark.

### 4.3 Comparison with Self-attention

To compare co-evolution attention with self-attention, a CoT model variant where the CoA module is replaced by the self-attention module is implemented (denoted as CoT-SA). Both CoT and CoT-SA are evaluated on the CASP14 and CAMEO benchmarks. As shown in Table 1, the CoT outperforms CoT-SA by 6.4%, 7.5%, 7.7%, and 6.8% *Precision@L* scores on four kinds of targets, indicating that CoA is much better than self-attention for this task. To further understand the co-evolution attention, the two attention matrices for both models are visualized. As shown in Figure 3 (a) and (c), the co-evolution attention patterns are much closer to the ground-truth contact map than that of self-attention, illustrating that CoA is more effective in extracting contact patterns.

### 4.4 Ablation Study

To evaluate the contribution of each model component, we set up ablative configurations for the co-evolution aggregation submodule, the enhancement submodule, and the self-attention module.

Experiments with various ablative configurations are conducted as listed in Table 3. For the co-evolution aggregation submodule, an *average pooling* is implemented as an alternative of *selective pooling*, where the features of different sequences are aggregated equally. The result shows that selective pooling obviously plays a critical role in the model, as the *Precision@L* score of the models with selective pooling on FM domains increased from 42.7% to 48.2% compared with that with average pooling.

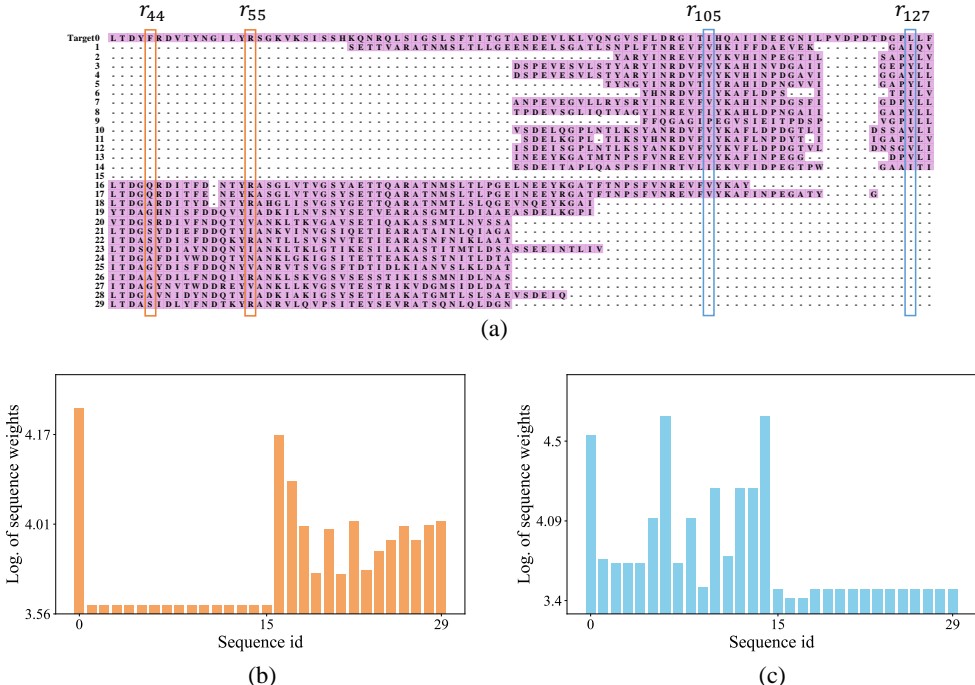

Figure 4: Learned sequence weights for `T1061-D1`. (a) Part of MSA for `T1061-D1` (residue 40-130), where the purple region represents aligned residues and '-' stands for a gap. (b) The sequence weight distribution for the pair $\langle r_{44}, r_{55} \rangle$. (c) The sequence weight distribution for the pair $\langle r_{105}, r_{127} \rangle$.

Figure 4 (a) is an MSA for the CASP14 target `T1061-D1`. Since the model is not sensitive to MSA sequence order, we rearrange the MSA for better visualization. Then we visualize the sequence weights CoT learned from the MSA for two selected residue pairs $\langle r_{44}, r_{55} \rangle$ and $\langle r_{105}, r_{127} \rangle$. Figure 4 (b) and (c) show that selective pooling assigns larger weights to more homologous sequences as expected. Meanwhile, the difference between two sequence weight distributions of the two pairs demonstrates that the model is able to learn pair-specific weights.

The *co-evolution enhancement* is a convolutional module to learn correlations among multiple residues, and the overall performance reduces significantly on FM domains from $48.2\%$ to $41.6\%$ when removing this module, as compared in Table 3. These results illustrate that the high-order residue interaction is very important for contact prediction, as the interaction might reflect some structure patterns (e.g, structure motifs) [14, 39]. Moreover, as illustrated in Figure 3, by applying co-evolution enhancement, long-range residue interactions can be better extracted.

By ablating the self-attention module, the overall model performance drops slightly, indicating that the CoA module is the main contributor to achieve the performance.

## 4.5   Model Analysis

**The Effect of MSA Quality**    The quality of predicted contacts is highly related to $M_{eff}$ [40], which is the number of the non-redundant sequence homologs in the MSA (sequences of more than $70\%$ identity is considered redundant). For the CASP14 FM domains, the correlation coefficients between the logarithm of $M_{eff}$ and the *Precision@L* scores is $0.63$, demonstrating that the performance of contact prediction is strongly correlated to MSA quality. CoT cannot predict the contact maps of some proteins very well (e.g., the *Precision@L* score of `T1093-D1` is only $20.6\%$). For most of these proteins, the $M_{eff}$ is as small as less than 20, indicating that low-quality MSAs are still a bottleneck for contact prediction.

**The Effect of Training Data**    Most methods generate the training data from the same data sources, i.e., public protein sequence databases (UniProt, Metagenome database) and structure databases

(PDB). However, different methods customize the data processing pipelines (e.g., MSA generation strategies), causing slightly different training data for the models. To study the effect of training data, we train another model CoT$^*$ on a smaller training dataset but of comparable size to that used by CopulaNet [2], i.e., PDB30. The performance of CoT$^*$ is slightly lower than CoT, e.g., $46.3\%$ vs. $48.2\%$ for *Precision@L* on the CASP14 FM domains and $66.1\%$ vs. $66.6\%$ on the CAMEO hard targets. But it still outperforms CopulaNet a lot, for example, $46.3\%$ vs. $38.2\%$ for *Precision@L* on CASP14 FM domains. These results demonstrate that the performance gain of CoT is not mainly achieved by the larger size of the training data.

## 5 Conclusion

We propose an attention-based architecture (CoT) to learn residue coevolutions from multiple sequence alignments (MSAs). As the core component of CoT, co-evolution attention (CoA) leverages the full information of an MSA to learn residue representations by treating coevolutionary patterns as attention. Moreover, it employs a selective pooling operation to mitigate the influence of non-homologous information. The experimental results on two rigorous benchmarks demonstrate the effectiveness of CoT.

On the other hand, the experimental results also reveal a failure case of CoT, i.e., CoT cannot accurately predict the contacts of proteins with low-depth MSAs. This is an issue shared by other existing approaches as well. How to address the issue caused by low-quality MSAs remains an open problem. We believe pretrained protein models may be a potential solution to it [26–30].

## Acknowledgements

We thank all the anonymous reviewers for their valuable comments. The work was supported in part with National Key R&D Program of China Grant 2017YFA0700800.

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
