# OpenReview forum: "Co-evolution Transformer for Protein Contact Prediction"
_NeurIPS.cc/2021/Conference — NeurIPS 2021 Poster_

### Official Review · Reviewer_WfQv · 2021-07-14

**Rating:** 6
**Confidence:** 3

**Summary:**

This manuscript describes a novel deep learning architecture for predicting protein contact maps. This is a well-studied problem for which deep models already provide the state of the art.  The innovation here is to use a particular type of Transformer architecture that models more dependencies than previous models.  The approach is well described, seems sensible, and gives good performance on a challenging benchmark. On the other hand, the empirical results are unsatisfying in some respects (outlined below), and the benchmarking suffers from a problem (different methods trained on different training sets).

**Main Review:**

The key idea of this work is that, in predicting co-evolutionary information for contact prediction, it is important not only to weight sequences non-uniformly and to capture dependencies between residues within each sequence, as has been done by previous model, but also to capture dependencies between sequences in the multiple sequence alignment (the "co-evolution enhancement" model).  The model architecture is well described in Figures 1-2 and Section 3, though I wish the architecture of CopulaNet had been explained in a bit more detail, so as to highlight more clearly which aspects of the proposed model are novel.

The text makes three primary claims (lines 173-179) about what makes this model novel: "1) COA leverages the global information from all the homologs instead of single homolog to derive the attention, which fits better to the residue co-evolution insight; 2) COA handles the non-homologous information naturally by the selective pooling operation during aggregation, providing a solution
to a widely existed but inevitable dilemma for MSAs; 3) COA enhances the co-evolution signal by
propagating information from the neighbors, making it easier to capture the high-order interactions
among multiple residues."   I would have liked to see each of these claims directly supported by empirical evidence in Section 4.  The results in Sections 4.3 and 4.4 go partway in this direction but do not directly test the above hypotheses.  For instance, 4.3 shows that the proposed co-evolution layers give better results overall than a standard self-attention layer but does not delve into the source of this improvement.

The results in Table 1 are problematic because the different methods are not trained on the same data.  I actually assumed that they were, at first, and only figured out for sure that they were not when I got to Section 4.5 (line 273).  The text should say this explicitly up front, since it makes the performance results reported in Table 1 less trustworthy.  We don't know whether the differences in performance are due to differences in the training set or due to modeling choices.  This is, in my opinion, a major problem with the reported results, though apparently it's a problem in the field as a whole ("... most state-of-the-art models are trained on their private training data, which are usually unavailable.").  The results on lines 275-279 go partway toward mitigating this problem.

line 220: "The improvement on FM/TBM domains (0:8%) is not that significant as the others (> 9:0%)." In fact, no statistical significance was assessed for any of these results. This should be relatively straightforward to do by, e.g., computing the performance measures separately per protein and then doing a statistical test (e.g., paired t-test) on those measures.

I could not make sense of these sentences: "Different groups use different data sources to build their MSAs. To eliminate the variance of different data sources, the most confident prediction of CoT on 12 MSAs from different databases with different search settings, denoted as CoTy, is selected as the final prediction."  It sounds like we are comparing models trained on different training sets.  How does the proposed strategy address this problem?

We are told that "Figure 4 (b) and (c) show that selective pooling assigns larger weights to more relevant sequences as expected," but I am not sure what "more relevant" means here.  The plot seem to show only that the weights are not uniform.

Figure 3 provides anecdotal evidence that including the co-evolution enhancement module improves prediction of long-range contacts. It would have been better to systematically measure this in a quantitative way.


**Time Spent Reviewing:**

1

---

> ### Author Response · Authors · 2021-08-10
> **Response to Reviewer WfQv**
>
> Thanks for your extremely valuable comments. Your concerns about the experimental results are addressed below.
>
> * Q1: *The model architecture is well described in Figures 1-2 and Section 3, though I wish the architecture of CopulaNet had been explained in a bit more detail, so as to highlight more clearly which aspects of the proposed model are novel.*
>
> We overview the architecture of CopulaNet and list the differences between the two models as below:
>
> CopulaNet[1] is a neural network model designed to learn co-evolution information directly from MSAs.
> It begins with a 1D ResNet encoder to generate residue embeddings, followed by an outer product operation over all sequences to derive residue co-evolution features on the averaged pair residue embeddings, and ends with a 2D ResNet to predict contacts from the co-evolution features.
>
> Compare to CopulaNet,
>
> 1) CoT derives the co-evolution patterns from all homologs, fitting better to the premise that residue co-evolutions reflect collective covariation patterns of numerous homologs, while CopulaNet learns residue representations from individual sequences independently.
>
> 2) CoT learns per-residue-pair weights to lower the influence of those irrelevant residue pairs, i.e. gaps and those from dissimilar subsequences, while CopulaNet assigns equal weights to the features from different homologs, ignoring the existence of non-homologous residues and gaps.
>
> 3) CoT provides a specific Transformer-based solution that could fit multiple dependencies of MSAs, while CopulaNet is a completely CNN-based architecture.
>
> * Q2: *How to directly support three primary claims of our model with empirical evidence.*
>
> Thanks for this question. Let us make some clarifications:
>
> 1) Claim 1: co-evolution attention (from all homologs) fits better to
> the residue co-evolution insight than vanilla self-attention (from a
> single homolog). The results in Table 1 empirically support this
> claim, where CoT (with co-evolution attention) outperforms CoT-SA
> (with self-attention) by 5.7\%, 3.8\%, 8.2\%, and 6.6\% Precision@L
> scores on the four kinds of targets. It is difficult to measure the source of improvement rigorously. Thus, we visualized the learned attention map in Figure 3, where the co-evolution attention patterns seem much closer to the true contact map than that of self-attention, illustrating CoA is more effective in extracting contact-related patterns.
>
> 2) Claim 2: the selective pooling operation could eliminate the non-homologous information in MSAs. The ablation study result in Table 3 shows that selective pooling outperforms average pooling by 4.8\% Precision@L score on CASP14-FM domains. We also visualized the learned per-residue-pair weights from CoT in Figure 4, where selective pooling assigns smaller weights to gaps (i.e., non-homologous residues) as expected.
>
> 3) Claim 3: propagating information from neighbors makes it easier to
> capture high-order residue interactions. We consider this point the
> strength of CoA over self-attention and this idea is not first used by
> CoA. Actually, most of the existing methods (e.g., RaptorX[2], AlphaFold[3], and CopulaNet) utilize a CNN module to model such high-order interactions, and we adopt this insight in our method, however, in the attention part. The ablation study result showed in Table 3  empirically proves the necessity of the enhancement submodule. Figure 3 shows that enhancing the co-evolution attention leads to better contact-related patterns.
>
> * Q3: *The results in Table 1 are problematic because the different methods are not trained on the same data.*
>
> By "the same data" we mean the same input data (i.e., MSAs)
> for the model. Most methods (including CoT) generate the training data
> from the same data sources, i.e., public protein sequence databases
> (UniProt, Metagenome database) and structure databases (PDB). However,
> different method customizes the data processing pipelines (e.g., MSA
> generation strategies) slightly differently, causing slightly
> different training data for the models. This slight difference
> generally has a negligible impact; the result in lines 275-279 demonstrates that different training data has very limited influence on the performance of CoT (46.3\% vs. 47.5\% Precision@L on FM domains).
>
> * Q4: *"The improvement on FM/TBM domains (0:8%) is not that significant as the others (> 9:0%)." In fact, no statistical significance was assessed for any of these results.*
>
> Thanks for the very constructive suggestion. Regarding that 14 FM/TBM samples are not enough to derive a statistical significance, we re-evaluated the CoT model and the CopulaNet on FM/TBM domains with 12 groups of MSAs (described in Section 4.1), obtaining 168 Precision@L scores for each method. The average Precision@L scores of CoT and CopulaNet are 59.5\% and 55.9\%, respectively.
>
> Assume the results follow a normal distribution,
> we conduct a Paired-samples t-test for the two groups of results, obtaining a t-statistic 3.02 and a p-value 0.003.
> Since the results may be not normally distributed,
> the Wilcoxon signed-rank test is further conducted, obtaining a t-statistic 3594.5 and a p-value 7.78e-6.
> These two results demonstrate that CoT is better than CopulaNet significantly in statistics.
>
> * Q5: *How to understand "Different groups use different data sources to build their MSAs. To eliminate the variance of different data sources, the most confident prediction of CoT on 12 MSAs from different databases with different search settings, denoted as CoTy, is selected as the final prediction." It sounds like we are comparing models trained on different training sets. How does the proposed strategy address this problem?*
>
> We have probably not made our points clearly. Let us further clarify:
> by "different groups use different data sources ...", we mean
> "different participants of CASP14 use different data sources
> ...". The CASP14 participants usually built multiple MSAs using
> different databases for a protein target and selected the one that
> worked the best.  Following this conventional strategy, during the
> inference phase, we also built 12 MSAs for each CASP14 target and
> reported the one that worked the best in Table 2. For the problem
> "different method uses different training data", we have discussed
> it in **Q3**.
>
> * Q6: *The meaning of "more relevant" in "Figure 4 (b) and (c) show that selective pooling assigns larger weights to more relevant sequences as expected."*
>
> "More relevant" refers to "more homologous with the target
> protein". The selective pooling operation learns to assign larger
> weights to more homologous/informative sequences instead of those
> non-homologous sequences and gaps.
>
> *Q7: *Figure 3 provides anecdotal evidence that including the co-evolution enhancement module improves prediction of long-range contacts. It would have been better to systematically measure this in a quantitative way.*
>
> We understand the reviewer's concern that Figure 3 is not rigorous
> enough to support the effectiveness claim of the co-evolution enhancement
> module. We did an ablation study about this module and reported the
> result in Table 3, where the Precision@L score is increased from
> 41.6\% to 47.5\% on FM domains with the co-evolution enhancement
> submodule, demonstrating the necessity of this module in a
> quantitative way. Inspired by previous works[1, 3, 4, 5], we also
> visualized these attention maps as Figure 3 to intuitively explain why the co-evolution works well.
>
> [1] Ju F, Zhu J, Shao B, et al. CopulaNet: Learning residue co-evolution directly from multiple sequence alignment for protein structure prediction[J]. Nature communications, 2021, 12(1): 1-9.
>
> [2] Wang S, Sun S, Li Z, et al. Accurate de novo prediction of protein contact map by ultra-deep learning model[J]. PLoS computational biology, 2017, 13(1): e1005324.
>
> [3] Senior A W, Evans R, Jumper J, et al. Improved protein structure prediction using potentials from deep learning[J]. Nature, 2020, 577(7792): 706-710.
>
> [4] Rao R, Meier J, Sercu T, et al. Transformer protein language models are unsupervised structure learners[C]//International Conference on Learning Representations. 2020.
>
> [5] Bhattacharya N, Thomas N, Rao R, et al. Single Layers of Attention Suffice to Predict Protein Contacts[J]. bioRxiv, 2020.

---

### Official Review · Reviewer_iKoC · 2021-07-14

**Rating:** 6
**Confidence:** 3

**Summary:**

The authors address an important problem in structural biology and obtain what looks like state-of-the-art performance.  The proposed method predicts inter-residue contacts of a protein based on its sequence presented as a multiple sequence alignment (MSA).  The method is innovative in the way it represents an MSA using attention on individual sequences within the MSA, avoiding the step of summarizing the MSA as a "profile" that summarizes amino acid composition in each position, a common way of representing MSAs.


**Limitations And Societal Impact:**

Yes.

**Main Review:**


Strengths:

- The idea of using attention to selectively focus on individual sequences within the MSA is innovative and interesting.  In their case study the authors demonstrate how this allows the method to focus on sequences for which no gap is present in a given position.

- State-of-the-art performance on a highly important problem in structural biology.

Weaknesses:

- I think the Co-evolution aggregation submodule (equations 7 and 8) and the co-evolution enhancement submodule could have been explained a bit more clearly.


Comments:

- If there is a gap in the original sequence in the MSA, that position  cannot be a part of any interactions.  It is not clear how this is addressed in the method.

- It is not clear how the MSA is provided as input.  Given my understanding of the model, I assume the input is a one hot encoding of each sequence in the MSA.  That step is also missing in Figure 1.

- Section 3.3 compares the expressivity of the proposed CoT layer with a simpler attention model over individual sequences.  However, in my opinion, a more natural comparison is with a model that summarizes the MSA by a sequence profile.

- It's not clear how the selected model parameters were chosen.  No model selection strategy was described.

Minor comments:

Please discuss the following method, as it also uses attention:
Chen, C., Wu, T., Guo, Z., & Cheng, J. (2021). Combination of deep neural network with attention mechanism enhances the explainability of protein contact prediction. Proteins: Structure, Function, and Bioinformatics, 89(6), 697-707.
I suspect it does not perform as well as the proposed method since its performance is similar to RaptorX.

- The following statement is incorrect:  "Transformer is a network architecture built on the self-attention mechanism" - transformers are constructed on the basis of attention rather than self-attention.



**Time Spent Reviewing:**

3.5

---

> ### Author Response · Authors · 2021-08-10
> **Response to Reviewer iKoC**
>
> Thanks for reviewing our paper and appreciating our idea. To address your concerns, we first summarize the design rationale of CoT, then we answer your questions.
>
> **Clarification for the Co-evolution Attention Module**
>
> Given an MSA (i.e., aligned protein sequences), the goal of the Co-evolution Attention Module is to leverage the whole MSA to derive pairwise inter-residue interaction features, namely residue co-evolutions.
> Intuitively, the residue co-evolutions are analogous to the covariance matrix in mathematics,
> depicting the correlations between any two residue positions (two columns in the MSA).
> Then, these features are used as attention maps to guide representation learning of each sequence in the MSA.
>
> We use the following two submodules to achieve this goal:
>
> 1) Given the MSA embedding ($K \times L \times C$, $K$: sequence number, $L$: sequence length, $C$: feature dimension), the **Co-evolution Aggregation submodule** first transforms each sequence embedding into a pairwise feature map, and then uses a selective pooling strategy to aggregate $K$ feature maps to a co-evolution feature map ($L \times L \times C$).
>
> 2) Given the co-evolution feature map, the **Co-evolution Enhancement submodule** extracts the high-order interactions among multiple residues by propagating information from residue neighbors.
> This operation is widely used to enhance the co-evolution signals in previous works[1, 2, 3].
>
> After these steps, the co-evolution feature map is projected into an attention map ($L \times L \times H$, $H$: number of attention head) to guide the representation learning of each sequence in the MSA.
>
> **Response to Comments**
>
> * Q1: *If there is a gap in the original sequence in the MSA, that position cannot be a part of any interactions. It is not clear how this is addressed in the method.*
>
> We fully agree with your insight that the gaps should not be a part of any interactions.
> In Co-evolution Transformer, the selective pooling strategy is
> proposed to learn the per-residue-pair weights so that to alleviate
> the influence of those pairs containing gaps as the reviewer
> mentioned. For comparison, most previous works (e.g., RaptorX[1],
> CopulaNet[2]) simply treat the gaps as an extra "word" in the
> vocabulary, ignoring that these gaps do introduce noises to the
> modeling of residue interactions. Both the empirical results and the
> visualization in section 4 demonstrate that our strategy can better handle this issue thus boosts the performance of contact prediction.
>
> * Q2: *It is not clear how the MSA is provided as input. That step is also missing in Figure 1.*
>
> We should have indeed made this clearer. We described this point in
> Appendix B.1 due to the page limit, and we will highlight it in the
> main text in the further revision.
> As we described in the appendix, the MSA is provided as input in the
> following way: inspired by CopulaNet[2], we represent one MSA as a collection of sequence pairs, where each pair consists of the target protein and another homologous protein. Each position in the sequence pair is encoded by a 41-dim binary vector. The first 20 dimensions are a one-hot encoding of the target protein residue; the other 21 dimensions are a one-hot encoding of the homologous protein residue, where the additional dimension denotes the **gaps**.
>
> * Q3: *Section 3.3 compares the expressivity of the proposed CoT layer with a simpler attention model over individual sequences. However, in my opinion, a more natural comparison is with a model that summarizes the MSA by a sequence profile.*
>
> Thanks for the constructive suggestion. A classical application of MSA profile is RaptorX[1], which takes both sequence profile and DCA-based features derived from MSAs as input. RaptorX is much stronger than just using a single sequence profile due to the information loss of the inter-residue correlation information within each sequence. This method has been taken as a baseline for CoT, and the empirical results in Table 1 show that CoT outperforms RaptorX 9.0\% Precision@L score on the CASP14-FM targets, demonstrating the strength of CoT.
>
> * Q4: *It's not clear how the selected model parameters were chosen. No model selection strategy was described.*
>
> Thanks for pointing this out. As we described in Appendix B.2, the parameters of the final model are selected according to the performance on the validation set, and a detailed comparison of different hyperparameter settings (e.g., the number of CoT layers and the embedding dimension) are summarized. We will emphasize this in the main text in the revision.
>
> * Q5: *Discussion about [Chen2021] "Combination of deep neural network with attention mechanism enhances the explainability of protein contact prediction"*
>
> We thank the reviewer for providing this related work. [Chen2021] did a solid exploration on incorporating the attention mechanism into contact prediction. [Chen2021] designed two attention modules to extract features from PSSM and PLM profiles and then used a ResNet to refine the inter-residue feature to predict protein contacts. The differences between CoT and [Chen2021] can be summarized as below:
>
> 1) The biggest difference is that [Chen2021] is a DCA-based method like RaptorX [2], adopting hand-crafted features as input to predict protein contacts. While Co-evolution Transformer extracts residue co-evolution features directly from MSAs. Previous works (e.g., CopulaNet) have demonstrated the limitations of the DCA-based methods, e.g. "The hand-crafted features only consider single-residue and pairwise statistics of the sequences, causing considerable information loss."
>
> 2) In addition, to derive residue co-evolution features, CoT is equipped with a dedicated attention module to jointly model multiple homologs as well as a selective pooling strategy to eliminate the influence of non-homologous information. These two components further contribute to the final performance.
>
> * Q6: *The following statement is incorrect: "Transformer is a network architecture built on the self-attention mechanism"*
>
> Thanks for pointing this out.
> We will revise it to "Transformer is a network architecture built on the attention mechanism".
>
> [1] Wang S, Sun S, Li Z, et al. Accurate de novo prediction of protein contact map by ultra-deep learning model[J]. PLoS computational biology, 2017, 13(1): e1005324.
>
> [2] Ju F, Zhu J, Shao B, et al. CopulaNet: Learning residue co-evolution directly from multiple sequence alignment for protein structure prediction[J]. Nature communications, 2021, 12(1): 1-9.
>
> [3] Senior A W, Evans R, Jumper J, et al. Improved protein structure prediction using potentials from deep learning[J]. Nature, 2020, 577(7792): 706-710.

---

### Official Review · Reviewer_qdTM · 2021-07-18

**Rating:** 7
**Confidence:** 3

**Summary:**

The authors present a model that predicts contact maps from multiple sequence alignments. The paper proposes a weighted sum of k, LxL attention maps (k=#sequences, L=sequence length), where a per-residue-pair weight is learned across all sequences in the MSA. Afterwards a convolutional layer is applied to refine the prediction. The authors show that the co-evolution transformer outperforms existing supervised contact prediction methods on CASP14.

**Limitations And Societal Impact:**

The authors have addressed the main limitation of their work, namely that the co-evolution transformer does not perform well on low-depth MSAs. Solving this problem is beyond the scope of the paper.

**Main Review:**

The idea of computing per-residue-pair weight across all sequences in the MSA is the main contribution of the paper. The general concept of attention maps for contacts and convolutional layers for refinement are well-known techniques in the field. The authors have clearly demonstrated the uniqueness and contribution of their idea through proper ablations. Related work is adequately cited.

The explanation of the co-evolution transformer is sound. One detail the authors should clarify in Line 161 (submodule 2) is when “the co-evolution map A is concatenated with A' from the previous CoT layer”, how is this handled in the first CoT layer when A’ does not yet exist? As a side question, why did the authors not additionally try a ResNet for the co-evolution enhancement? The expectation is that this would perform at least as well as the conv-linear-softmax block proposed in the paper.

There seems to be little comparison of Co-evolution Transformer performance on the different MSA generation methods (different databases). This is an interesting topic, and it is not clear why the authors did not mention it in the main paper.

Most claims are well supported, but two are suspect.
Line 97: [Pretraining-based methods] are still at an early stage thus cannot achieve comparable performance to the SOTA approaches currently.
Line 104: [MSA Transformer] it still cannot well extract the co-evolution information due to the non-homologous subsequences inevitably introduced by the generated MSAs.

These claims don't seem to be correct. Re the claim on line 97, MSA Transformer (Rao, et al. 2021) shows SOTA contact prediction on CASP13 free modeling targets relative to trRosetta. As such, it’s not clear why this model wasn’t included in the comparisons. Re the claim on line 104, Rao, et al, 2021 show the model extracts co-evolution information from MSAs through experiments with unsupervised contact prediction. These claims should be revised, and the paper would be strengthened by a comparison.

50.4 Precision @ L on CASP14 free modeling targets compared to 41.8 is compelling. When comparing to the known methods (RaptorX, TrRosetta, CopulaNet), the authors should clarify whether the precision scores from the baseline methods are based on predicted contact maps OR the contact maps of predicted 3D structures.

As the current field stands, there is no single accepted way on how deep learning models should process MSAs. This paper presents a unique approach that merits consideration from the broader community.

Additional notes: Please define how you determine “the most confident prediction” [line 226]

---------------------------------
EDIT: See discussion around "Training data and comparisons". Authors have agreed to address the temporal split in the revision, which will strengthen the comparison to other methods.


**Time Spent Reviewing:**

4

---

> ### Author Response · Authors · 2021-08-10
> **Response to Reviewer qdTM**
>
> Thanks for reviewing our paper and providing constructive comments. Your questions are answered as follows:
>
> * Q1: *How does the first CoT layer handle the concatenation of $A$ and $A'$?*
>
> Thanks for pointing this out. We should have made this clearer.
> For the first layer when $A'$ doesn't exist, we simply assign $A'$ an all-zero tensor.
>
> * Q2: *Why did the authors not try a ResNet for the co-evolution enhancement?*
>
> Thanks for your suggestion. We realized that we did not state it
> clearly. The convolution module for the co-evolution enhancement does
> adopt a ResNet and we will clarify it in the revision.
>
> * Q3: *There seems to be little comparison of Co-evolution Transformer performance on the different MSA generation methods (different databases). This is an interesting topic, and it is not clear why the authors did not mention it in the main paper.*
>
> This is indeed a good suggestion and we should have included such a
> comparison. The Precision@L scores of CASP14-FM domains and CAMEO
> targets on the four MSAs generated from different databases are summarized as follows.
>
> | Method | Benchmark  | UniRef30 | UniRef90 | BFD30 | MGnify90 |
> | :-----: | :-----: | :-----: | :-----: | :-----: | :-----: |
> | CoT (ours) | CASP14-FM| 30.0 | 30.0 | 47.5 | 37.2 |
> | CoT (ours) | CAMEO | 67.6 | 66.7 | 66.4 | 58.3 |
> | CopulaNet | CASP14-FM | 25.1 | 25.4 | 38.5  | 31.8 |
> | CopulaNet | CAMEO | 58.0 | 57.0 | 56.5 | 49.2 |
>
> 1) The performance of Co-evolution Transformer for different MSAs
> varies a lot. For FM domains, CoT performs the best on the BFD30
> generated MSA. For the CAMEO targets, CoT has similar performance scores on the first three generated MSAs.
> A possible reason is that the FM domains are harder than the CAMEO
> targets due to the fact that fewer homologs for FM domains exist in most databases (e.g., UniRef).
> Therefore, CoT is able to obtain a better result on an extremely large database BFD30 for FM domains.
>
> 2) We also conduct the same experiments on the CopulaNet model, which shows a similar phenomenon as CoT does.
>
> We will add more details in the revised manuscript as discussed above.
>
> * Q4: *Two claims about [Pretraining-based methods] and [MSA Transformer] are suspect.*
>
> Thanks for pointing this out and we found out that it is a
> misunderstanding mainly due to poor organization of sentences, so let us make it clearer here.
> For the claim on Line97, the original intention is "[Pretraining-based methods for single protein sequences] are still at an early stage ...". For the claim on Line 104, we think "[MSA Transformer] did solid work on learning co-evolution information from unlabeled MSAs. In contrast, we leverage a specific co-evolution attention module and a selective pooling strategy to model MSAs in a supervised way."
> These two methods are quite different in design and methodology. We will revise the claims in the manuscript.
>
> * Q5: *It’s not clear why this model wasn’t included in the comparisons. ..., and the paper would be strengthened by a comparison.*
>
> [MSA Transformer] wasn't included in the comparison for two reasons:
> 1) it is an unsupervised model pretrained on a large amount of extra
> sequence data;
>
> 2) we cannot find the supervised fine-tuning
> code/script in the [MSA Transformer] public repository (we have indeed
> sent emails to the authors a few months ago but got no replies and we cannot reproduce the results on the CASP14 benchmark.)
>
> Nevertheless, we rerun the CoT model on the CASP13-FM dataset to compare with it, obtaining a 65.0\% Precision@L score, which is better than 57.1\% reported in [MSA Transformer]. For a fair comparison, we have filtered protein sequences similar to CASP13 targets (sequence identity more than 25\%) in the training set.
>
> * Q6: *Whether the precision scores from the baseline methods are based on predicted contact maps OR the contact maps of predicted 3D structures.*
>
> All precision scores are calculated based on "predicted contact
> maps". We will make it clearer in the revised manuscript.
>
> * Q7: *Definition of "the most confident prediction"*
>
> Thanks for your suggestion. The final prediction for a residue pair is
> a probability value denoting whether the contact exists or not. The
> most confident prediction refers to those pairs with the highest
> probability scores.

---

> > ### Comment · Reviewer_qdTM · 2021-09-01
> > **Training data and comparisons**
> >
> > In my review I seem to have missed an important point raised by reviewer WfQv about the comparisons using different training data. This undermines the comparisons with other methods. I don't think the author response adequately addresses this.
> >
> > The temporal holdout for CASP14 appears to be invalid. According to section 4.1 Benchmark and Dataset, the first CASP14 target was released on May 18, 2020. The model is trained using a validation set of structures from Apr. 1, 2020 to Dec. 25, 2020. i.e. overlapping with CASP14. As a result my evaluation of the paper has changed and I am lowering my score. Please correct me if I am wrong.

---

> > > ### Author Response · Authors · 2021-09-02
> > > **Response to Reviewer qdTM - II**
> > >
> > > * *In my review I seem to have missed an important point raised by reviewer WfQv about the comparisons using different training data. This undermines the comparisons with other methods. I don't think the author response adequately addresses this.*
> > >
> > > For different training data, as our response to Reviewer WfQv: 1) both protein structures and sequences are collected from the same public databases; 2) different data preprocessing pipelines will cause slightly different input for the models, which leads to a negligible influence on the final performance, typically about 1% for the precision score (CASP14-FM).
> > >
> > > * *The temporal holdout for CASP14 appears to be invalid. According to section 4.1 Benchmark and Dataset, the first CASP14 target was released on May 18, 2020. The model is trained using a validation set of structures from Apr. 1, 2020 to Dec. 25, 2020. i.e. overlapping with CASP14.*
> > >
> > > For the temporal validation split, let us make some further clarifications to address the reviewer's concern on overlapping between our validation set and the CASP14 test set.
> > >
> > > The short answer is, there is no overlap between our validation set and the CASP14 test set with respect to the structure labels, since all data samples for the CASP14 targets were released to PDB after Dec. 25, 2020. In fact, this potential issue was purposely avoided in our experiments.
> > >
> > > By "the first CASP14 target was released on May 18, 2020.", we mean that the protein **sequence** of the target was released to CASP contest participants. During this period, no **protein structures (labels)** are publicly available; the structures were not released to PDB before Dec. 25, 2020. Therefore, there are no overlapping structures between our validation set and the CASP14 test set.
> > >
> > > Moreover, to avoid introducing similar protein sequences with the CASP14 targets to our validation set, we had filtered out the sequences with sequence similarity larger than 25% (a common threshold to keep a rigorous blind test, as used in RaptorX[1]) with the CASP14 targets. The protein lists of our training and validation sets were provided in our supplementary material along with the submitted manuscript.
> > >
> > > With the experimental settings described above, the potential data leakage issue for our experimental evaluation was completely avoided.
> > >
> > > [1] Wang S, Sun S, Li Z, et al. Accurate de novo prediction of protein contact map by ultra-deep learning model[J]. PLoS computational biology, 2017, 13(1): e1005324.

---

> > > > ### Comment · Reviewer_qdTM · 2021-09-02
> > > > **Re**
> > > >
> > > > Thank you very much for the additional clarifications. This work contributes interesting ideas around using attention for modeling proteins, a topic of considerable interest.
> > > >
> > > > The choices made around data used to train the models make the comparison to other methods difficult to evaluate. The standard practice for comparisons using CASP test sets is to use a temporal split, taking care to ensure that the validation set is consistent with this split. Otherwise the comparison is not completely fair as model selection and tuning is performed with protein structures that would not have been available to the other methods.
> > > >
> > > > Recent work including RaptorX (Xu 2019), trRosetta (Yang et al. 2020), CopulaNet (Ju et al. 2021), all use temporal splits for their comparisons on CASP. In these papers both the training and validation data are temporally held out from the test set.

---

> > > > > ### Author Response · Authors · 2021-09-03
> > > > > **Response to Reviewer qdTM - III**
> > > > >
> > > > > Thank you very much for your response. Excluding all the **similar** sequences to the targets is a rigorous and agreed pipeline to prepare the protein sequence data, this is even more rigorous than the time splits in terms of modeling, since it not only avoids all the potential leakage but also removes the potential structure templates for reference. Thus, we followed this pipeline, which is also performed in RaptorX (Wang et al. 2017) and updated RaptorX (Xu et al. 2021), and had used a strict threshold (25\%) to avoiding introducing any similar structure labels with CASP14 into the validation set. This rigorous pipeline is always followed by the previous work while the time splits strategy is not:
> > > > >
> > > > > *- For the CASP11 benchmark, RaptorX (Wang et al. 2017) used PDB25 created in **February 2015** as their training set, while the first CASP11 target is released before **May. 1, 2014**. For the CASP13 benchmark, trRosetta (Yang et al. 2020) used Uniclust30 (**version2018_08**) to generate MSAs for training and evaluation, while the first CASP13 target is released in **May. 1, 2018**. For the CASP13 benchmark, RaptorX (Xu et al. 2021, the latest version of RaptorX) used structure labels between **March 2018** and **1. January 2020** as the validation set.*
> > > > >
> > > > > Nevertheless, we fully understand your concern about the time splits and agree that an additional comparison is necessary. Since the quantitative comparison with time splits cannot be provided now due to the limited rebuttal time, we will complement this experiment in the revised manuscript. To our experience, the proposed method is not sensitive to different choices of validation sets. Thus the conclusions from the reported results still hold most likely.
> > > > >
> > > > > [Xu et al. 2021] Xu J, Mcpartlon M, Li J. Improved protein structure prediction by deep learning irrespective of co-evolution information[J]. Nature Machine Intelligence, 2021: 1-9.

---

> > > > > > ### Comment · Reviewer_qdTM · 2021-09-03
> > > > > > **Re: Response to Reviewer qdTM - III**
> > > > > >
> > > > > > There are two types of data that need to be considered: the version of the structure database, and the version of the sequence database. My comment was around the temporal split on the structures; the works mentioned in that comment follow a temporal split. The authors rightly point out that there are some deviations in the literature. And because this issue only affects the validation set, its impact is more limited. I'm editing the review to increase the score because this is a complicated issue that also affects other work in the field and the authors have agreed to address it in a revision.

---

### Decision · Program_Chairs · 2021-09-27

**Decision:**

Accept (Poster)

**Comment:**

This paper wishes to address the issue of protein contact prediction problem, State of the art approach uses hand-crafted features through Multiple Sequence Alignments(MSAs). This paper proposes a Transformer architecture to automatically identify features suitable for Protein Contact Prediction problem. All the referees were unanimous that while the ideas are novel but their comparison with the state of the art
is problematic. We hope that the authors can address these concerns in the final manuscript.